# BALANCED MULTIMODAL LEARNING: AN UNIDIRECTIONAL DYNAMIC INTERACTION PERSPECTIVE

## ABSTRACT

Multimodal learning typically utilizes multimodal joint loss to integrate different modalities and enhance model performance. However, this joint learning strategy can induce modality imbalance, where strong modalities overwhelm weaker ones and limit exploitation of individual information from each modality and the inter-modality interaction information. Existing strategies such as dynamic loss weighting, auxiliary objectives and gradient modulation mitigate modality imbalance based on joint loss. These methods remain fundamentally reactive, detecting and correcting imbalance after it arises, while leaving the competitive nature of the joint loss untouched. This limitation drives us to explore an alternative approach that avoids reliance on the joint loss, aiming to foster more effective modality interactions and to better exploit both per-modality information and inter-modality complementarity. In this paper, we introduce Unidirectional Dynamic Interaction (UDI), a proactive sequential training strategy that replaces conventional joint optimization. UDI first trains the anchor modality to convergence, then uses its learned representations to guide the other modality via unsupervised loss. Furthermore, the dynamic adjustment of modality interactions allows the model to adapt to the task at hand, ensuring that each modality contributes optimally. By decoupling modality optimization and enabling directed information flow, UDI prevents domination by any single modality and fosters effective cross-modal feature learning. Our experimental results demonstrate that UDI outperforms existing methods in handling modality imbalance, leading to performance improvement in multimodal learning tasks. (The code will be published.)

## 1 INTRODUCTION

In recent years, multimodal learning has gained significant attention due to its potential to leverage information from multiple modalities, such as images, text, and audio, to improve model performance across various tasks Baltrušaitis et al. (2019); Han et al. (2023); Liang et al. (2022b; 2024). However, recent studies have highlighted a persistent challenge: in many settings, one modality tends to dominate the learning process, while others remain underexploited, a problem known as modality imbalance Peng et al. (2022); Wang et al. (2020); Wu et al. (2022). When certain modalities overshadow their counterparts, the model fails to capture the full spectrum of information, leading to suboptimal fusion and degraded performance. To mitigate this imbalance, researchers have explored a variety of techniques. One line of strategies seek to rebalance gradients, slowing down updates from dominant modalities so that weaker modalities can catch up during training Fan et al. (2023); Li et al. (2023); Peng et al. (2022). Another group of methods accelerate the training of weaker modalities by discarding features from the dominant modality during training Wei et al. (2025a); Yang et al. (2025). Researchers have also explored fusion module methods that aim to achieve modality balance by adjusting how different modalities are integrated Wu et al. (2022); Zhang et al. (2024). Moreover, a set of approaches introduce auxiliary loss to enhance the training of weaker modalities, ensuring more balanced learning across modalities Du et al. (2023); Ma et al. (2023); Yang et al. (2024).

While existing methods mitigate modality imbalance to some extent, they still depend on a single joint loss to facilitate modality interaction. This approach suffers from two intrinsic flaws. To validate these observations, we run a simple experiment comparing three training schemes on CREAM-D including (1) Decouple: each modality is trained independently, fusion result is ob-

tained by mean-weighted averaging of unimodal outputs; (2) MMPareto Wei & Hu (2024): a representative joint-loss baseline that applies a Pareto-style reconciliation between unimodal and multimodal losses; (3) UDI (Ours): a method for achieving directional interaction between modalities based on decoupled information flow. For all schemes we report three evaluations: audio (unimodal) accuracy, video (unimodal) accuracy and fusion accuracy, using the same experiment setups to ensure a fair comparison. From Fig. 1, we can see that under MMPareto, the unimodal accuracies of both audio and video are lower than those of the decoupled unimodal training baseline, indicating modality imbalance persists. A uniform joint objective cannot adapt to each modality's distinct informativeness. Implicit modality competition remains hidden in the joint objective, so stronger modalities continue to dominate. Meanwhile, the fusion performance of MMPareto is also lower than that of decoupled training and our UDI, suggesting that joint loss based modality interaction remains shallow and the complementary information in weaker modalities is not fully leveraged due to the joint loss's inability to reconcile conflicting gradients from different modalities.

To overcome these flaws, we introduce Unidirectional Dynamic Interaction (UDI), which decouples modality optimization from the outset by training modalities sequentially. First select the modality branch with the highest standalone performance on the downstream task as the anchor and fully train the anchor without interference. Then freeze its parameters before guiding, thereby eliminating the hidden competition inherent in joint loss. Knowledge from the anchor is then used to guide other modality branches through a unified unsupervised objective that both enforces consistency with the anchor's soft predictions and encourages the other modality to uncover specific features, ensuring that complemen-

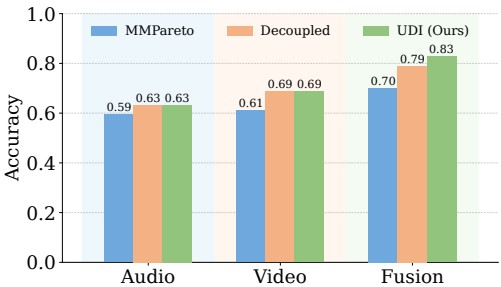

Figure 1: Comparison of audio, video and fusion accuracies under three training schemes.

tary information in weaker streams is actively leveraged rather than buried in tangled gradients. Moreover, a simple dynamic controller continuously rebalances the consistency and complementary terms throughout training. UDI ensures no single modality can dominate and all modalities to contribute effectively to multimodal integration.

The main contributions of our work are summarized as follows: (1) We introduce a novel training strategy that decouples information flow by sequentially optimizing modalities. we first select the highest performing branch as an anchor and train it to convergence. Then use anchor to guide weaker streams, thereby eliminating hidden competition and directly addressing modality imbalance. (2) We propose a unified unsupervised distillation framework combining a consistency loss and a complementary loss. All under a dynamic controller that adaptively rebalances these objectives based on task demands, ensuring truly balanced, deep cross-modal interactions. (3) Extensive experiments on benchmark datasets show that UDI achieves superior performance compared to existing baselines, highlighting its effectiveness in addressing modality imbalance and improving multimodal learning.

## 2 RELATED WORK

### 2.1 IMBALANCED MULTIMODAL LEARNING

Various methods have been proposed to address the issue of modality imbalance in multimodal learning. These approaches generally focus on adjusting how modalities contribute to the model's learning process to ensure that weaker modalities are not neglected. Techniques such as fine-grained evaluation and methods like OPM Wei et al. (2025a) aim to dynamically adjust the contributions of different modalities during training. Additionally, strategies like MLA Zhang et al. (2024) and Greedy Wu et al. (2022) modify how features from different modalities are fused to improve the interaction between them. Other methods, such as CML Ma et al. (2023) adjust the loss functions to balance the learning between modalities by emphasizing weaker modalities through various loss constraints. Optimization-based methods like OGM Peng et al. (2022) and AGM Li et al. (2023) adjust gradient magnitudes to prevent dominant modalities from overshadowing weaker ones, while

techniques like PMR Fan et al. (2023) and Relearning Wei et al. (2025b) address modality imbalance by adjusting gradients or reinitializing model weights. These strategies help alleviate modality imbalance but often fall short in fully addressing the complex interactions between modalities.

## 2.2 MUTUAL INFORMATION ESTIMATION

Mutual Information (MI) has become a crucial tool in various domains, especially for regularization and controlling dependencies between variables. In many cases, MI is used to constrain the independence between variables, offering valuable insights into the structure of the data. For instance, Kim et al. Hjelm et al. (2019) utilizes MI to perform unsupervised representation learning by maximizing the mutual information between the input and output of a deep neural network. Similarly, Kim et al. Kim & Mnih (2018) uses MI to learn disentangled representations by encouraging independence between the components of the learned representations. The application of MI minimization has also gained attention in the field of disentangled representation learning. Studies like those by Cheng et al. (2020) introduce efficient methods for estimating MI, such as a contrastive log-ratio upper bound, which provides an approximation for scenarios where only sample data from the joint distribution is available. Moreover, Dunion et al. Dunion et al. (2023) focuses on minimizing conditional MI between representations to enhance model generalization, particularly in tasks involving correlated features or shifts in data distributions. Despite its widespread use, mutual information estimation has not been explored in depth for specific tasks like modality imbalance or for learning disentangled representations in this context. This gap presents an opportunity to extend MI-based approaches to improve the handling of modality imbalance and enhance the overall performance and robustness of multimodal learning tasks.

## 3 METHODOLOGY

In this section, we introduce our proposed multimodal unidirectional dynamic interaction approach. Our method consists of two sequential steps. First, we avoid the hidden competition by decoupling training for each modality branch and objectively selecting the branch with the highest downstream performance as the anchor. Second, we freeze the anchor's parameters and use its learned representations to guide the remaining modalities via a unified unsupervised loss that combines consistency and complementary terms. Our method proactively eliminates modality imbalance by decoupling optimization. Simultaneously creating a directed information flow from the anchor to other modality branches enable deeper, more balanced cross-modal feature learning. The overall workflow is illustrated in Fig. 2.

### 3.1 ANCHOR MODALITY LEARNING

We first select the highest-performing branch as the anchor; if multiple branches exhibit similar validation performance, we choose the one with the lower predictive entropy, i.e. smaller $H(\hat{y}) = -\sum_c \hat{y}_c \log \hat{y}_c$ (its corresponding modality is denoted by $a$). Concretely, we train each modality branch to convergence in isolation and pick the one with the best standalone validation accuracy. Its parameters are frozen before subsequent guidance. For completeness, we also reports results when alternative branches are used as the anchor in Appendix A.7. The following is the formalization of the training process (trimodal and more see Appendix A.2).

Given a dataset $X = \{x_i\}_{i=1}^N$ consisting of $N$ samples, where each sample contains $M$ different modalities $x_i = \{x_i^1, x_i^2, \ldots, x_i^M\}$. The ground truth labels are represented as $Y = \{y_i \mid y_i \in \{0,1\}^c\}_{i=1}^N$, where $c$ denotes the number of category labels.

For each modality $m$, a dedicated deep encoder extracts an intermediate feature representation. We denote the feature extractor for modality $m$ as $\psi^m(\cdot)$ with parameters $\theta^m$. Therefore, for the $i$-th sample, the feature vector is given by:

$$f_i^m = \psi^m(x_i^m; \theta^m), \quad f_i^m \in \mathbb{R}^{d_m}, \tag{1}$$

where $d_m$ denotes the dimensionality of the feature vector extracted from the $m$-th modality.

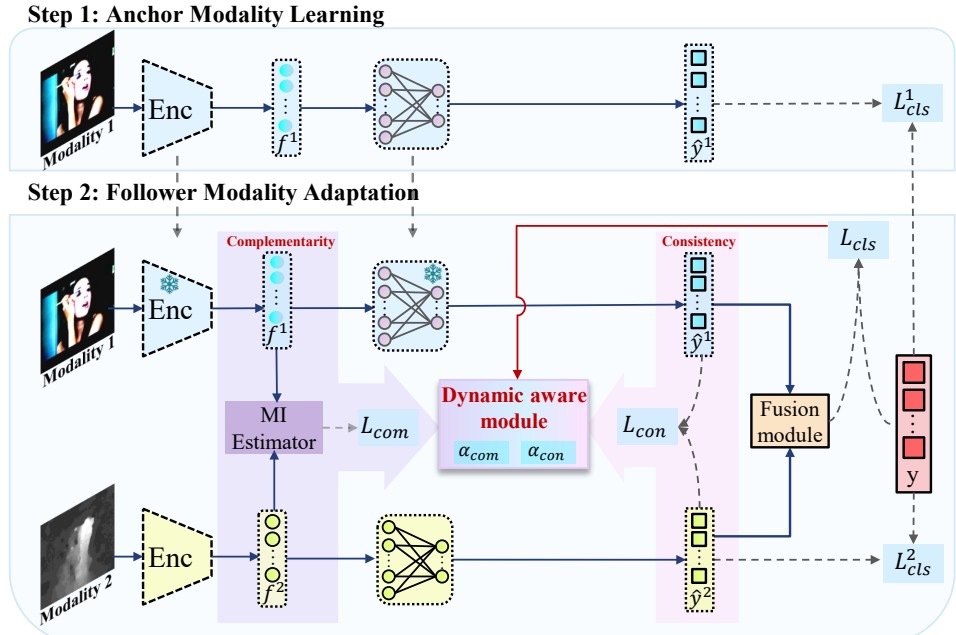

Figure 2: The whole framework of the proposed model.

Once intermediate features from the different modalities are obtained, the features are then processed by a fully-connected decision layer to produce the final class probabilities:

$$\hat{y}_i^m = \text{softmax}\left(W^m f_i^m + b^m\right), \tag{2}$$

where $W^m \in \mathbb{R}^{d_m \times c}$ and $b^m \in \mathbb{R}^c$ are the weights and bias of modality $m$ respectively. The overall loss for training anchor $a$ is expressed as:

$$L_{\text{total}}^a = L_{\text{cls}}^a = -\frac{1}{N} \sum_{i=1}^{N} y_i^\top \log \hat{y}_i^a, \tag{3}$$

where $\hat{y}_i^a$ denotes the prediction results based solely on anchor modality $a$.

## 3.2 FOLLOWER MODALITY ADAPTATION

After obtaining a well-trained anchor, its parameters are fixed (i.e., frozen). The pretrained features $f_i^a$ and the corresponding decision outputs $\hat{y}^a$ are then used to guide the training of other modalities (e.g., modality $m \neq a$). In this phase, in addition to the classification loss $L_{\text{cls}}^m$, we create a directed information flow from the anchor to the followers by introducing two unsupervised loss terms: a consistency loss $L_{\text{con}}^m$ and a complementary loss $L_{\text{com}}^m$.

The consistency loss encourages the decision outputs of the follower modality $\hat{y}^m$ and anchor modality $\hat{y}^a$ to be similar. We implement this with the Jensen–Shannon divergence:

$$\text{JS}(\hat{y}^a, \hat{y}^m) = \frac{1}{2}\text{KL}\left(\hat{y}^a \,\|\, H\right) + \frac{1}{2}\text{KL}\left(\hat{y}^m \,\|\, H\right), \tag{4}$$

where the Kullback–Leibler divergence $\text{KL}(P\|Q) = \sum_k P(k) \log \frac{P(k)}{Q(k)}$ and $H = \frac{1}{2}\left(\hat{y}^a + \hat{y}^m\right)$ is the mixture distribution. Thus, the consistency loss of follower modality $m$ is formulated as:

$$L_{\text{con}}^m = \frac{1}{2} \sum_{i=1}^{N} \left[ \text{KL}\left(\hat{y}_i^a \,\Big\|\, \frac{\hat{y}_i^a + \hat{y}_i^m}{2}\right) + \text{KL}\left(\hat{y}_i^m \,\Big\|\, \frac{\hat{y}_i^a + \hat{y}_i^m}{2}\right) \right]. \tag{5}$$

To encourage the follower to learn modality-specific (complementary) features, we minimize the mutual information(MI) between the anchor and the follower representations $f^a$ and $f^m$. MI quan-

tifies the amount of shared information between two random variables and is defined as:

$$I(f^a; f^m) = \int p(f^a, f^m) \log \frac{p(f^a, f^m)}{p(f^a)p(f^m)} df^a \, df^m = \mathbb{E}_{p(f^a, f^m)} \left[ \log \frac{p(f^a, f^m)}{p(f^a)p(f^m)} \right]. \quad (6)$$

However, directly computing $I(f^a; f^m)$ for high-dimensional data is intractable. To address this, we estimate a tight upper bound of the MI. First, we define the MI upper bound estimator as (proof in Appendix A.1):

$$\hat{I}(f^a; f^m) = \mathbb{E}_{p(f^a, f^m)} \left[ \log p(f^m|f^a) \right] - \mathbb{E}_{p(f^a)} \mathbb{E}_{p(f^m)} \left[ \log p(f^m|f^a) \right]. \quad (7)$$

Since the true conditional distribution $p(f^m|f^a)$ is not directly available, we approximate it with a variational distribution $q_\phi(f^m|f^a)$ parameterized by a lightweight network $Q_\phi$ (e.g. $q_\phi(f^m|f^a) = Q_\phi(f^m, f^a)$). In a discretized form, the variational MI estimator becomes:

$$\hat{I}_v(f^a; f^m) = \frac{1}{N^2} \sum_{i=1}^{N} \sum_{j=1}^{N} \left[ \log q_\phi(f_i^m|f_i^a) - \log q_\phi(f_j^m|f_i^a) \right]. \quad (8)$$

To make this upper bound as tight as possible, we minimize the KL divergence between the true conditional distribution and its variational approximation:

$$\min_\phi \text{KL}\left( p(f^m|f^a) \,\|\, q_\phi(f^m|f^a) \right) = \min_\phi \mathbb{E}_{p(f^a, f^m)} \left[ \log p(f^m|f^a) \right] - \mathbb{E}_{p(f^a, f^m)} \left[ \log q_\phi(f^m|f^a) \right]. \quad (9)$$

Since the first term in the KL divergence is independent of $\phi$, this reduces to minimizing the negative log-likelihood:

$$L_{\text{MI}} = -\frac{1}{N} \sum_{i=1}^{N} \log Q_\phi(f_i^a, f_i^m). \quad (10)$$

Minimizing $L_{\text{MI}}$ trains the MI estimator $Q_\phi^m$ to accurately approximate the conditional distribution $p(f^m|f^a)$. Once $Q_\phi^m$ is well-trained, we use the variational MI upper bound $\hat{I}_v(f^a; f^m)$ as our complementary loss:

$$L_{\text{com}}^m = \hat{I}_v(f^a; f^m). \quad (11)$$

By minimizing $L_{\text{com}}^m$ during the training of follower modality $m$, we effectively reduce the mutual information between $f^a$ and $f^m$, thereby promoting complementary feature representations. The overall loss for training follower modality $m$ is as follows:

$$L_{\text{total}}^m = L_{\text{cls}}^m + \alpha_{\text{con}} \cdot L_{\text{con}}^m + \alpha_{\text{com}} \cdot L_{\text{com}}^m, \quad (12)$$

where $\alpha_{\text{con}}$ and $\alpha_{\text{com}}$ are adaptive weights that balance the consistency and complementary losses for follower $m$. This formulation creates a directed information flow from the anchor modality $a$ to the follower modality $m$, providing sufficient unidirectional interaction.

### 3.3 DYNAMIC AWARE MECHANISM

To adaptively modulate the interaction between anchor and follower modality $m$, we introduce a dynamic aware strategy that measures alignment between the task gradient (from the multimodal classification loss) and the gradients of the two unsupervised losses. The gradients are denoted as:

$$g_{\text{cls}} = \nabla_\theta L_{\text{cls}}, \quad g_{\text{con}} = \nabla_\theta L_{\text{con}}^m, \quad g_{\text{com}} = \nabla_\theta L_{\text{com}}^m, \quad (13)$$

where $\theta$ denotes the model parameters. We quantify directional agreement by computing the inner product of gradients restricted to the set of shared parameters:

$$\xi_{\text{con}} = \sum_{k \in \mathcal{K}_{\text{con}}} \left( g_{\text{cls}}[k] \cdot g_{\text{con}}[k] \right), \quad \xi_{\text{com}} = \sum_{k \in \mathcal{K}_{\text{com}}} \left( g_{\text{cls}}[k] \cdot g_{\text{com}}[k] \right), \quad (14)$$

where $\mathcal{K}_{\text{con}} = g_{\text{cls}}.keys() \cap g_{\text{con}}.keys()$ and $\mathcal{K}_{\text{com}} = g_{\text{cls}}.keys() \cap g_{\text{com}}.keys()$ are the sets of shared parameters. These scalar values measure how much each unsupervised loss pushes parameters in the same direction as the classification objective. We then apply a ReLU function to keep only positive contributions:

$$\tilde{\xi}_{\text{con}} = \text{ReLU}(\xi_{\text{con}}), \quad \tilde{\xi}_{\text{com}} = \text{ReLU}(\xi_{\text{com}}). \quad (15)$$

Finally, we normalize these nonnegative alignments to obtain adaptive weights:

$$\alpha_{\mathrm{con}} = \frac{\tilde{\xi}_{\mathrm{con}}}{\tilde{\xi}_{\mathrm{con}} + \tilde{\xi}_{\mathrm{com}} + \epsilon}, \quad \alpha_{\mathrm{com}} = \frac{\tilde{\xi}_{\mathrm{com}}}{\tilde{\xi}_{\mathrm{con}} + \tilde{\xi}_{\mathrm{com}} + \epsilon}, \tag{16}$$

where a small $\epsilon > 0$ prevents division by zero. Intuitively, these weights emphasize unsupervised components whose gradient directions agree with the task objective and attenuate those that conflict. In experiment, we compute gradients on a single selected mini-batch and update $\alpha_{\mathrm{con}}$, $\alpha_{\mathrm{com}}$ per epoch. This update schedule lets the controller adapt over time while keeping overhead low.

In summary, our framework first fully optimizes the selected anchor. Then guides followers learning using a unified unsupervised objective which employs the dynamic aware mechanism to adaptively balance the unsupervised losses, yielding a robust, directed information flow.

We admit that UDI introduces extra training-time cost compared with a single joint-run. However, in many applications training time is less critical than inference latency and model footprint. Crucially, the extra components used by UDI (the anchor selection and the dynamic controller) are only needed during training and do not incur additional cost at deployment. In addition, we adopt several strategies to reduce the training time cost: (1) parallelize or shorten the pre-training time of each modality in anchor selection; (2) the dynamic controller update $\alpha$ infrequently (one selected mini-batch per epoch). The complete algorithm is provided in Appendix A.3.

## 4 EXPERIMENTS

### 4.1 EXPERIMENTAL SETUP

**Datasets.** We select six widely used datasets for experimental evaluation: CREMA-D Cao et al. (2014), Kinetics-Sounds (KS) Arandjelovic & Zisserman (2017), Colored-and-gray-MNIST (CGM-NIST) Kim et al. (2019), UCF101 Soomro et al. (2012), Food-101 Wang et al. (2015), and CMU-MOSEI Bagher Zadeh et al. (2018). These datasets cover a diverse range of modality combinations, including audio-visual, gray-color, text-visual, optical flow-RGB, and audio-visual-text. Detailed descriptions of each dataset are provided in the Appendix A.4.

**Compared Methods.** We compare our proposed method against a comprehensive set of methods, including traditional multi-modal learning (MML) methods and the latest techniques specifically designed for handling imbalanced data. Compared methods comprise the conventional summation (Sum) method as well as several state-of-the-art unbalanced approaches, namely CML Ma et al. (2023), GBlending Wang et al. (2020), MMPareto Wei & Hu (2024), LFM Yang et al. (2024), OGM Peng et al. (2022), AGM Li et al. (2023), PMR Fan et al. (2023), Relearning Wei et al. (2025b), ReconBoost Hua et al. (2024), MLA Zhang et al. (2024), OPM Wei et al. (2025a), Greedy Wu et al. (2022), and Modality-valuation Wei et al. (2024). For detailed descriptions of each baseline, please refer to the Appendix A.5.

**Implementation Details.** Details on network architectures, training hyperparameters and fusion strategy for each dataset can be found in Appendix A.6.

### 4.2 COMPARISON WITH IMBALANCED METHODS

To verify the superiority of the proposed method, we present the results on the various datasets, as shown in Table 1. In Table 1, Unimodal-1 and Unimodal-2 represent the audio and video modalities for the audio-visual datasets, the gray and color modalities for CGMNIST, and the RGB and optical flow modalities for UCF101. For Food-101, Unimodal-1 and Unimodal-2 refer to the text and image modalities, respectively. In the case of CMU-MOSEI, Unimodal-1 and Unimodal-2 represent audio and video modalities, while Unimodal-3 exclusively represents the text modality, which is only considered in the CMU-MOSEI dataset.

From Table 1, the following conclusions can be drawn: (1) Our approach outperforms all baselines in terms of accuracy (ACC) and F1 score in most datasets. This highlights the effectiveness of our decoupling-based directed information flow design: by decoupling optimization and creating a one-way guidance from an anchor to followers, UDI better exploits both unimodal information and inter-modal complementarities. (2) In Food-101, our method achieved an accuracy of 92.89% and an F1

Table 1: Comparison of different methods on various datasets.

| Method | CREMA-D | | Kinetics-Sounds | | CGMNIST | | UCF101 | | Food-101 | | CMU-MOSEI | |
|---|---|---|---|---|---|---|---|---|---|---|---|---|
| | ACC | F1 | ACC | F1 | ACC | F1 | ACC | F1 | ACC | F1 | ACC | F1 |
| Unimodal-1 | 63.17% | 63.59% | 54.54% | 53.7% | **99.32%** | **99.32%** | 78.35% | 77.55% | 86.28% | 86.32% | 71.33% | 43.05% |
| Unimodal-2 | 68.68% | 68.61% | 55.82% | 54.62% | 71.69% | 71.18% | 70.08% | 69.76% | 65.71% | 65.73% | 71.23% | 49.56% |
| Unimodal-3 | - | - | - | - | - | - | - | - | - | - | 81.09% | 74.22% |
| Sum | 66.40% | 66.82% | 65.57% | 64.58% | 64.07% | 63.68% | 81.79% | 81.31% | 90.32% | 90.29% | 78.96% | 71.37% |
| CML | 69.18% | 69.57% | 67.56% | 67.22% | - | - | 84.74% | 84.28% | 92.70% | 92.66% | 79.69% | 73.16% |
| GBlending | 71.59% | 71.72% | 68.82% | 66.43% | - | - | 85.01% | 84.50% | 92.56% | 92.50% | 79.64% | 73.29% |
| MMPareto | 79.97% | 80.57% | 70.13% | 70.18% | 81.88% | 81.69% | 85.30% | 84.89% | 92.82% | 92.77% | 81.18% | 74.64% |
| LFM | 70.02% | 69.55% | 66.37% | 66.02% | 97.77% | 97.75% | 84.95% | 84.35% | 92.58% | 92.54% | 79.90% | 71.60% |
| OGM | 67.76% | 68.02% | 67.04% | 66.95% | 66.40% | 66.26% | 82.07% | 81.30% | 91.81% | 91.77% | 80.45% | 73.61% |
| AGM | 71.59% | 72.11% | 66.62% | 65.88% | 67.64% | - | 81.70% | 80.89% | 91.89% | 91.84% | 79.86% | 71.89% |
| PMR | 67.19% | 67.20% | 67.11% | 66.87% | 78.50% | - | 81.93% | 81.48% | 92.10% | 92.04% | 79.88% | 72.09% |
| Relearning | 71.02% | 71.46% | 65.92% | 65.48% | - | - | 82.87% | 82.15% | 91.68% | 91.63% | 78.65% | 70.02% |
| ReconBoost | 74.01% | 74.52% | 68.38% | 67.68% | - | - | 82.89% | 82.26% | 92.47% | 92.44% | 81.01% | 74.03% |
| MLA | 72.30% | 72.66% | 69.05% | 68.75% | 71.40% | - | 85.38% | 84.84% | **93.14%** | **93.09%** | 78.65% | 70.02% |
| OPM | 68.75% | 69.00% | 66.89% | 66.44% | 76.11% | 76.05% | 85.28% | 83.79% | 93.08% | 93.04% | 79.95% | 72.83% |
| Greedy | 66.48% | 66.54% | 66.82% | 66.53% | 91.01% | - | - | - | 73.80% | 71.21% | - | - |
| Modality-valuation | 75.85% | 76.68% | 68.01% | 68.03% | 68.56% | - | 85.25% | 84.69% | 92.20% | 92.15% | 79.84% | 72.99% |
| Ours | **82.80%** | **83.19%** | **74.68%** | **73.86%** | 99.12% | 99.11% | **86.33%** | **85.97%** | 92.89% | 92.90% | **81.84%** | **75.77%** |
| | ±0.45% | ±0.42% | ±0.32% | ±0.28% | ±0.03% | ±0.03% | ±0.15% | ±0.12% | ±0.03% | ±0.04% | ±0.01% | ±0.01% |

Table 2: Results of ablation study on various datasets.

| Method | | CREMA-D | | Kinetics-Sounds | | CGMNIST | | UCF101 | | Food-101 | | CMU-MOSEI | |
|---|---|---|---|---|---|---|---|---|---|---|---|---|---|
| DO | DI | ACC | F1 | ACC | F1 | ACC | F1 | ACC | F1 | ACC | F1 | ACC | F1 |
| ✗ | ✗ | 66.40% | 66.82% | 65.57% | 64.58% | 64.07% | 63.68% | 81.79% | 81.31% | 90.32% | 90.29% | 78.96% | 71.37% |
| ✗ | ✓ | 75.27% | 75.26% | 68.42% | 67.36% | 98.47% | 98.46% | 83.74% | 83.29% | 92.35% | 92.32% | 81.46% | 75.67% |
| ✓ | ✗ | 78.90% | 79.25% | 73.25% | 72.46% | 97.87% | 97.85% | 85.96% | 85.63% | 92.86% | 92.85% | 80.49% | 74.47% |
| ✓ | ✓ | **82.80%** | **83.19%** | **74.68%** | **73.86%** | **99.12%** | **99.11%** | **86.33%** | **85.97%** | **92.89%** | **92.90%** | **81.84%** | **75.77%** |

score of 92.9%. While this is among the top-performing methods, MLA and OPM achieved slightly higher results. This could be attributed to the relatively mild modality imbalance, and both image and text modalities provide important, complementary information. As a result, Food-101 may benefit from carefully designed joint-loss fusion. (3) In CGMNIST, the color modality is generated by adding color biases to the gray modality and contributes little new information. As a result, multimodal learning approaches often struggle to improve performance with the addition of the color modality. UDI demonstrates a significant advantage in this context. The decoupling-based directed information flow design allows the model to effectively learn from the anchor modality (gray) while prevent the follower modality from degrading performance. In conclusion, this ability to focus on the anchor modality while still benefiting from follower modalities highlights the effectiveness and adaptability of our method in addressing modality imbalance. We also reports results when alternative branches are used as the anchor in Appendix A.7.

## 4.3 ABLATION STUDY

In this section, we present the results of an ablation study to evaluate the contributions of the two key components of our method: decoupled optimization (DO) and dynamic interaction (DI). The results are summarized in Table 2. From Table 2, we can draw the following conclusions: (1) **Decoupled optimization only.** Training each modality independently without interaction or guidance between modalities yields clear improvements over the baseline on most datasets. This demonstrates that decoupling modality optimization removes the hidden competition introduced by a joint loss and prevents stronger streams from suppressing weaker ones. (2) **Dynamic interaction only.** Applying the dynamic controller to reweight unsupervised objectives based on gradient directions also improves performance over the baseline. This shows that dynamically selecting unsupervised tasks that agree with the classification gradient promotes more effective modality interactions. (3) **Decoupled optimization + Dynamic interaction.** Combining both components achieves the best results across all datasets. Decoupling provides a high-quality modality's representations. Dynamic controller evaluates the task alignment of each unsupervised objective and adaptively reweights them. together they enable balanced, task-appropriate, and deeper cross-modal interactions, yielding the strongest mitigation of modality imbalance.

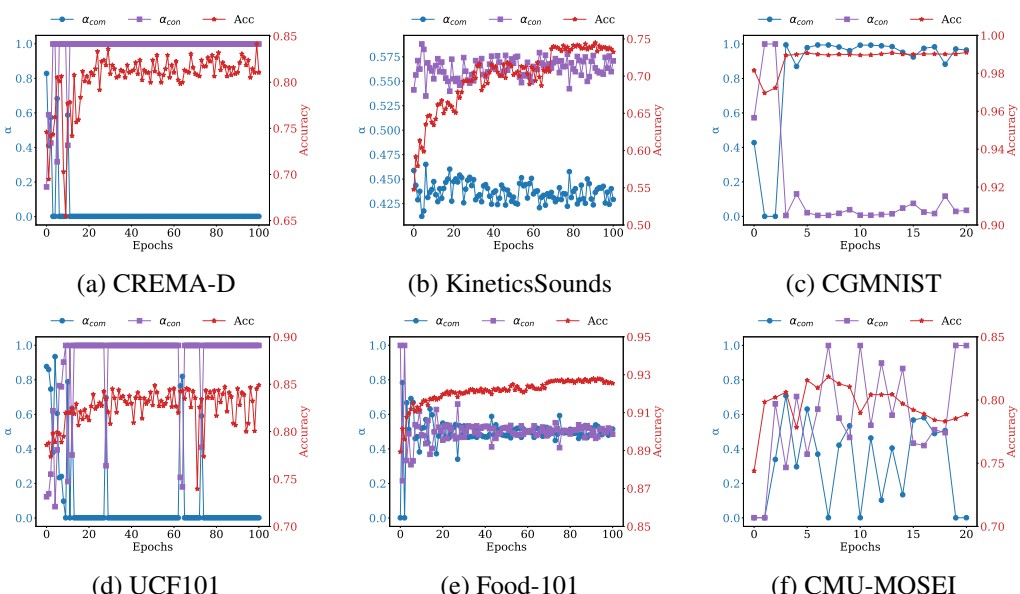

Figure 3: Loss weight and accuracy change across epochs for different datasets.

Table 3: Performance of $\alpha_{con}$ and $\alpha_{com}$ on various datasets.

| Method | | CREMA-D | | Kinetics-Sounds | | CGMNIST | | UCF101 | | Food101 | | CMU-MOSEI | |
|---|---|---|---|---|---|---|---|---|---|---|---|---|---|
| $\alpha_{con}$ | $\alpha_{com}$ | ACC | F1 | ACC | F1 | ACC | F1 | ACC | F1 | ACC | F1 | ACC | F1 |
| 0 | 1 | 81.59% | 81.97% | 73.71% | 72.68% | 99.02% | 99.02% | 85.83% | 85.47% | 92.38% | 92.39% | 81.48% | **76.22%** |
| 1 | 0 | 82.39% | 82.75% | 74.49% | 73.6% | 96.37% | 96.35% | 86.23% | 85.8% | 92.71% | 92.72% | 81.54% | 75.85% |
| 0 | 0 | 78.90% | 79.25% | 73.25% | 72.46% | 97.87% | 97.85% | 85.96% | 85.63% | 92.86% | 92.85% | 80.49% | 74.47% |
| | Ours | **82.80%** | **83.19%** | **74.68%** | **73.86%** | **99.12%** | **99.11%** | **86.33%** | **85.97%** | **92.89%** | **92.90%** | **81.84%** | 75.77% |

## 4.4 ANALYSIS OF LOSS WEIGHT

In this section, we analyze how the dynamic controller balances the follower's alignment to the anchor versus encouraging follower-specific (complementary) features during follower training. The per-epoch loss weight and accuracy curves of follower training on different datasets are plotted in Fig. 3. From Fig. 3, we observe that the dynamic controller considers different objectives in different datasets, which can be divided into the following three cases: (1) For CGMNIST, the controller favors the complementary term as training proceeds. Initially, since the follower has not been fully trained, the controller preserves a balance between alignment and exploration. But as training stabilizes, the model benefits more from extracting follower-specific features; consequently the complementary weight rises and stabilizes while accuracy improves. (2) In the case of CREMA-D and UCF101, the controller progressively increases the consistency weight, indicating that follower's alignment with the anchor is more helpful for classification task: the two modalities share strongly overlapping, task-relevant information, so reducing their prediction disparity improves fusion. Note that accuracy exhibits larger fluctuations on UCF101 likely caused by noisy optical-flow estimates, so stronger alignment also helps enhance the joint decision in practice. (3) For KineticsSounds, Food-101 and CMU-MOSEI the controller maintains a more balanced weighting between consistency and complementarity throughout follower training, suggesting that both aligning to the anchor and discovering follower-specific features are important.

To further validate the role of the two unsupervised terms, we fix the loss weights $\alpha_{con}$ and $\alpha_{com}$ to four configurations including (0,1) complementary-only, (1,0) consistency-only, (0,0) no unsupervised terms, and our adaptive schedule. We present the experimental results in Table 3. From the results, we observe the following: (1) Disabling both unsupervised losses ((0,0)) generally degrades performance relative to configurations that include at least one unsupervised term, confirming that the auxiliary objectives are beneficial for modality interaction. (2) The unsupervised objective that

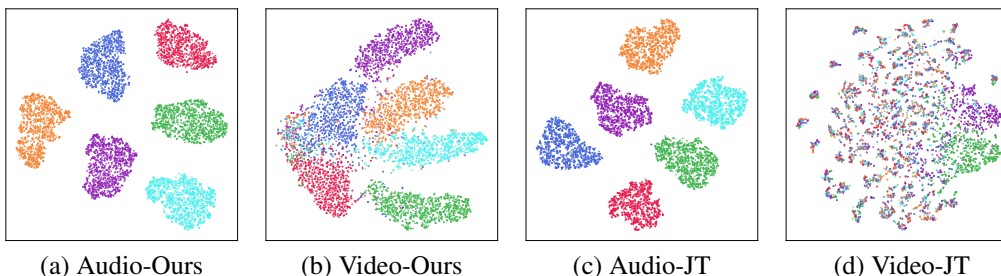

|(a) Audio-Ours|(b) Video-Ours|(c) Audio-JT|(d) Video-JT|

Figure 4: The visualization of the modality-specific feature by t-SNE van der Maaten & Hinton (2008) on CREMA-D dataset. The categories are indicated in different colors. JT denotes for joint training.

most benefits modality interaction varies across datasets: On CGMNIST, model benefit more from emphasizing the complementary term ((0,1) outperforms (1,0)). By contrast, model prefers the consistency objective ((1,0) outperforms (0,1)) on CREMA-D and UCF101. These are consistent with the observations in the previous section. (3) Our adaptive scheme achieves the best performance in all datasets, validating the effectiveness and robustness of task-dependent, dynamic weighting.

### 4.5 VISUALIZED REPRESENTATION ANALYSIS

To further illustrate the impact of our approach on unimodal feature learning, we project the representations of each modality into two-dimensional space using t-SNE van der Maaten & Hinton (2008), as shown in Fig. 4 We compare our Unidirectional Dynamic Interaction (UDI) method against a standard joint-training baseline. For the audio modality, both UDI and the joint-training baseline achieve similarly well-separated clusters, indicating that each can effectively capture the distinct characteristics of audio inputs. However, for the video modality, UDI produces markedly clearer separation between classes compared to the joint-training baseline. This enhanced separability demonstrates that UDI's unidirectional interaction prevents the dominant modality from monopolizing the learning process, allowing the weaker modality to develop stronger, more discriminative representations.

As discussed in Liang et al. (2022a), the modality gap measures the degree of separation between different modalities in a shared feature space, with a larger gap often correlating with improved performance. To evaluate the impact of dynamic interaction (DI), we visualize the modality distances for our UDI method both with and without DI on the CREMA-D dataset. In Fig.5 (Appendix), UDI with DI exhibits a substantially larger modality gap compared to UDI without DI. This increased gap demonstrates that the dynamic adjustment strategy effectively balances complementary and consistency losses, enabling the model to learn more discriminative representations and achieves higher accuracy, highlighting the effectiveness of our DI. In summary, our visual analysis confirms that UDI not only effectively mitigates modality imbalance but also learns more discriminative representations across modalities, leading to higher accuracy in classification tasks.

## 5 CONCLUSION

In this paper, we introduced a novel approach Unidirectional Dynamic Interaction (UDI) to address the challenges of modality imbalance in multimodal learning. UDI achieves decoupled modal interaction by first selecting and fully training a high-performing branch as an anchor, then establishing a directed information flow from that anchor to follower modalities via unsupervised distillation. The distillation objective combines a consistency term and a complementary term. We employ a dynamic controller adaptively balances these terms during training. Our experiments demonstrate that UDI effectively mitigates modality imbalance, leading to improved model performance in various multimodal tasks. Unlike existing methods that rely on joint loss function and thus reactively detect and correct modality imbalance, we provide a new perspective for balanced multimodal learning: UDI proactively eliminates imbalance by decoupling optimization and uses unsupervised losses to strengthen modality interactions.

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

# A  APPENDIX

In the Appendix:

- **A.1.** We prove the upper bound of the mutual information estimator.
- **A.2.** We describe the extension to tri-modal and multi-modal training.
- **A.3.** The overall algorithm of UDI.
- **A.4.** We give detailed descriptions of the datasets.
- **A.5.** We describe the basic information of all baselines.
- **A.6.** We explain the implementation details.
- **A.7.** We analyze the anchor selection.
- **A.8.** We present supplementary figure.

## A.1  PROOF OF THE UPPER BOUND OF MI

In this section, we demonstrate that the MI estimator $\hat{I}(x;y)$ serves as an upper bound for the true mutual information $I(x;y)$. $\hat{I}(x;y)$ and $I(x;y)$ are defined as:

$$\hat{I}(x;y) = \mathbb{E}_{p(x,y)}\big[\log p(y|x)\big] - \mathbb{E}_{p(x)}\mathbb{E}_{p(y)}\big[\log p(y|x)\big] \tag{17}$$

$$I(x;y) = \mathbb{E}_{p(x,y)}\left[\log \frac{p(x,y)}{p(x)p(y)}\right] = \mathbb{E}_{p(x,y)}\big[\log p(y|x)\big] - \mathbb{E}_{p(x,y)}\big[\log p(y)\big]. \tag{18}$$

We begin by considering the difference between the estimator and the true MI:

$$\begin{aligned}
\hat{I}(x;y) - I(x;y) &= \mathbb{E}_{p(x,y)}\big[\log p(y|x)\big] - \mathbb{E}_{p(x)}\mathbb{E}_{p(y)}\big[\log p(y|x)\big] \\
&\quad - \mathbb{E}_{p(x,y)}\big[\log p(y|x)\big] + \mathbb{E}_{p(x,y)}\big[\log p(y)\big] \\
&= \mathbb{E}_{p(x,y)}\big[\log p(y)\big] - \mathbb{E}_{p(x)}\mathbb{E}_{p(y)}\big[\log p(y|x)\big].
\end{aligned} \tag{19}$$

Since $\log p(y)$ is independent of $x$, we have:

$$\mathbb{E}_{p(x,y)}\big[\log p(y)\big] = \mathbb{E}_{p(y)}\big[\log p(y)\big]. \tag{20}$$

Thus, the difference simplifies to:

$$\hat{I}(x;y) - I(x;y) = \mathbb{E}_{p(y)}\big[\log p(y) - \mathbb{E}_{p(x)}[\log p(y|x)]\big]. \tag{21}$$

Now, observe that the marginal distribution $p(y)$ can be written as:

$$p(y) = \int p(y|x)p(x)\,dx = \mathbb{E}_{p(x)}\big[p(y|x)\big]. \tag{22}$$

Given that the logarithm is a concave function, Jensen's inequality yields:

$$\log p(y) = \log \mathbb{E}_{p(x)}\big[p(y|x)\big] \geq \mathbb{E}_{p(x)}\big[\log p(y|x)\big]. \tag{23}$$

This inequality implies:

$$\hat{I}(x;y) - I(x;y) = \mathbb{E}_{p(y)}\big[\log p(y) - \mathbb{E}_{p(x)}[\log p(y|x)]\big] \geq 0. \tag{24}$$

Therefore, we conclude that:

$$\hat{I}(x;y) \geq I(x;y). \tag{25}$$

Equality holds if and only if $\log p(y|x)$ is constant with respect to $x$, meaning that $p(y|x)$ does not vary with $x$ (i.e., $x$ and $y$ are independent). This completes the proof that $\hat{I}(x;y)$ is indeed an upper bound on $I(x;y)$.

## A.2 THE TRAINING PROCESS OF TRIMODAL AND MULTIMODAL

We first determine the single-modality performance order on a validation set and denote the modalities in descending order of performance as $a, b, c$. Modalities $a$ and $b$ are trained as described in the main text. After this stage we obtain their learned representations $f^a$, $f^b$ and fused output $\hat{y}^{ab}$. During the subsequent training of modality $c$ we use the previously obtained $(f^a, f^b, \hat{y}^{ab})$ to guide the learning of modality $c$. The difference from bimodal training is that the unsupervised loss $L_{\text{con}}$ and $L_{\text{com}}$ becomes:

$$L_{\text{con}}^c = \text{JS}(\hat{y}^{ab}, \hat{y}^c), \quad L_{\text{com}}^c = \tfrac{1}{2}(\hat{I}_v(f^a; f^c) + \hat{I}_v(f^b; f^c)). \tag{26}$$

Aligning the output of modality $c$ with the fused output $\hat{y}^{ab}$ helps the weaker modality $c$ produce predictions that are coherent with the stronger fused output $\hat{y}^{ab}$, improving robustness and reducing conflicting decisions. We can explicitly encourage modality $c$ to learn modality-specific features by minimizing mutual information between $f^c$ and $f^a$, $f^b$. The overall loss for training follower modality $c$ is as follows:

$$L_{\text{total}}^c = L_{\text{cls}}^c + \alpha_{\text{con}}^c \cdot L_{\text{con}}^c + \alpha_{\text{com}}^c \cdot L_{\text{com}}^c. \tag{27}$$

The progressive scheme naturally generalizes to more than three modalities: after training the first $m$ modalities and their fusion, the training of $(m+1)$-th modality is guided using the set of features and fused outputs produced by the first $m$ branches. The $(m+1)$-th training objective is formed by the consistency and complementary terms with the new branch's own supervised loss.

## A.3 ALGORITHM

---

**Algorithm 1** Unidirectional Dynamic Interaction Training Procedure

---

1: **Input:** Training dataset $\{(x_i^1, x_i^2, \ldots, x_i^M, y_i)\}_{i=1}^N$, epochs $\{E^m\}_{m=1}^M$
2: **Output:** Trained parameters $\{\theta^m\}_{m=1}^M$ and MI estimator parameters $\theta^{\text{MI}}$
3: **for** $m = 1$ to $M$ **do**
4:   **if** $m = 1$ **then**
5:     **for** epoch = 1 to $E^1$ **do**
6:       **for** each minibatch B from modality 1's paired data **do**
7:         Compute loss: $L_{\text{total}}^1 = -\frac{1}{|B|} \sum_{i \in B} y_i^\top \log \hat{y}_i^1$ and update $\theta^1$
8:       **end for**
9:     **end for**
10:    Freeze $\theta^1$
11:   **else**
12:     **for** epoch = 1 to $E^m$ **do**
13:       **for** each minibatch $B$ from modality $m$'s paired data **do**
14:         Compute $L_{\text{MI}}$ and update $\theta^{\text{MI}}$
15:         Compute $L_{\text{cls}}, L_{\text{cls}}^m, L_{\text{con}}^m, L_{\text{com}}^m$
16:         Compute gradients $g_{\text{cls}}, g_{\text{con}}, g_{\text{com}}$
17:         Compute positive gradient sums $\tilde{\xi}_{\text{con}}, \tilde{\xi}_{\text{com}}$
18:         Set adaptive weights $\alpha_{\text{con}}, \alpha_{\text{com}}$
19:         Compute loss $L_{\text{total}}^m = L_{\text{cls}}^m + \alpha_{\text{con}} \cdot L_{\text{con}}^m + \alpha_{\text{com}} \cdot L_{\text{com}}^m$ and update $\theta^m$
20:       **end for**
21:     **end for**
22:   **end if**
23: **end for**
24: **return** learned parameters $\{\theta^m\}_{m=1}^M$

---

## A.4 DATASETS

CREMA-D Cao et al. (2014) is an emotion recognition dataset featuring audio and visual modalities with six common emotions. It is divided into 6,698 clips training and 744 clips test set. Kinetic-Sounds (KS) Arandjelovic & Zisserman (2017) is an action recognition dataset that utilizes audio

and video modalities across 31 action classes selected from the Kinetics dataset, comprising approximately 19,000 10-second video clips. It is divided into 15,000 clips training, 1,900 clips validation set and 1,900 clips test set. Colored-and-gray-MNIST Kim et al. (2019) is a synthetic digit recognition dataset based on MNIST where each instance includes both a grayscale and a monochromatic colored image representing 10 digit classes (0–9). UCF101 Soomro et al. (2012) is an action recognition dataset with RGB and optical flow modalities covering 101 human action categories, divided into a 9,537-sample training set and a 3,783-sample test set according to the original setting. Food-101 Wang et al. (2015) is a large-scale multimodal dataset for food recognition and recipe analysis, containing image and text modalities over 101 food categories with about 100,000 recipe entries, each represented by one image and corresponding textual information. CMU-MOSEI Bagher Zadeh et al. (2018) is a multimodal sentiment and emotion recognition dataset that integrates audio, visual, and textual modalities, annotated with sentiment scores (ranging from –3 to +3) and six basic emotions.

## A.5 DETAILS OF BASELINES

**Summation** is a straightforward approach where the outputs from each modality are combined by summing them together. It does not prioritize any specific modality but treats each modality equally in contributing to the final prediction. This method serves as a baseline to compare against more sophisticated approaches that handle modality imbalance or interaction more explicitly.

**CML** Ma et al. (2023) proposes a regularization technique that ensures the model's predictive confidence does not increase when a modality is removed. It calibrates the confidence of multimodal predictions, improving the model's robustness and consistency across different modalities.

**GBlending** Wang et al. (2020) addresses training difficulties in multimodal classification by calculating the overfitting-to-generalization ratio (OGR) and dynamically adjusting gradient weights to minimize overfitting while improving generalization.

**MMPareto** Wei & Hu (2024) addresses gradient conflicts in multimodal learning caused by task difficulty disparities. It ensures that the gradient direction aligns with all learning objectives and adjusts the gradient magnitude to enhance generalization.

**LFM** Yang et al. (2024) promotes multimodal classification by dynamically learning modality gaps. It combines unsupervised contrastive learning with supervised multimodal learning using two strategies—heuristic-based and learning-based—to maximize the synergy between both approaches.

**OGM** Peng et al. (2022) dynamically monitors the contribution of each modality to the learning objective and adjusts gradients accordingly. By adding Gaussian noise for enhanced generalization, OGM-GE improves multimodal performance and can be integrated into existing multimodal models.

**AGM** Li et al. (2023) enhances model performance by using a Shapley value-based attribution method to isolate unimodal responses and adjusting the backpropagation signals for each modality. This helps modulate the training process and quantifies modality competition.

**PMR** Fan et al. (2023) introduces a prototype-based rebalancing strategy to address modality imbalance. It uses Prototype Cross-Entropy (PCE) loss to accelerate the clustering of slow-learning modalities and Prototype Entropy Regularization (PER) to penalize dominant modalities during the early stages of training, reducing their suppression of weaker modalities.

**Relearning** Wei et al. (2025b) dynamically adjusts unimodal encoder training based on each modality's learning status, as determined by the separability of its unimodal representation space. By soft-resetting encoder parameters, it prevents overemphasis on underrepresented modalities and enhances the training of weak modalities, achieving a balanced multimodal learning process.

**ReconBoost** Hua et al. (2024) tackles modality competition by alternating updates for each modality. It uses KL divergence-based coordination to update modality learners, improving overall performance by correcting errors in other modalities. It also introduces memory consolidation and global correction strategies.

**MLA** Zhang et al. (2024) breaks down traditional joint optimization into alternating unimodal optimization strategies to solve modality imbalance. It incorporates a gradient modification mechanism

to avoid modality forgetting and employs a dynamic fusion mechanism during testing to integrate multimodal information effectively.

**OPM** Wei et al. (2025a) together with On-the-fly Gradient Modulation (OGM), adjusts the optimization process in multimodal learning by dynamically changing modality weights during training. It reduces the impact of dominant modalities by modifying features in the feed-forward stage and gradients during backpropagation, improving the learning of suppressed modalities.

**Greedy** Wu et al. (2022) solves the greedy learning problem in multimodal deep neural networks by balancing the learning speeds of different modalities. It uses a proxy indicator to measure the learning progress across modalities, thereby improving model performance and generalization.

**Modality-valuation** Wei et al. (2024) introduces a sample-level modality valuation indicator that assesses the contribution of each modality to individual samples. By using Shapley values, it enhances the learning of under-contributing modalities, improving multimodal cooperation and overall model performance.

## A.6 IMPLEMENTATION DETAILS

To ensure a fair comparison across methods, we standardize the experimental settings for each dataset. For CREMA-D, we use ResNet18 as the backbone for audio-video tasks. Audio clips are converted into 257×299 spectrograms, while video segments are processed by extracting at 1 fps followed by uniform selection of 2 frames as visual inputs. For Kinetics-Sounds, ResNet18 similarly serves as the backbone, with audio transformed into 257×1004 spectrograms and video frames extracted at 1 fps followed by uniform sampling of 3 frames per clip. For CGMNIST, we adopt a specialized encoder containing four convolutional layers and one average pooling layer Fan et al. (2023), requiring no additional preprocessing. For UCF101, both RGB frames and optical flow sequences are processed by ResNet18. 3 frames are uniformly sampled from each clip as visual inputs and optical flow sequences generated by computing horizontal and vertical components stacked as [u,v] maps, where 3 flow frames are uniformly selected. For Food-101, visual features are extracted via a pre-trained ResNet18 with images resized to 256×256, and recipe texts are encoded by a pre-trained BERT model with captions truncated to a maximum of 40 characters. For CMU-MOSEI, we utilize a Transformer-based architecture for its three modalities without preprocessing. All models are trained in PyTorch on an NVIDIA RTX 4090 GPU using SGD with momentum = 0.9 and weight decay = 1e-4, learning rates of 1e-2 for CREMA-D and Kinetics Sounds and 1e-3 for Food-101 and UCF-101, a batch size of 64, summation fusion for two-modality datasets. We found that mean-weighted fusion on CMU-MOSEI yielded only 74% accuracy and it well below the text unimodal accuracy of 81%, indicating it failed to exploit cross-modal complementarity; therefore we adopt concatenation fusion for CMU-MOSEI.

## A.7 ANALYSIS OF ANCHOR SELECTION

Table 4: Performance with different anchor on various datasets.

| Anchor | CREMA-D | | Kinetics-Sounds | | CGMNIST | | UCF101 | | Food101 | | CMU-MOSEI | |
|---|---|---|---|---|---|---|---|---|---|---|---|---|
| | ACC | F1 | ACC | F1 | ACC | F1 | ACC | F1 | ACC | F1 | ACC | F1 |
| Unimodal-1 | 81.18% | 81.51% | 74.68% | 73.86% | 99.12% | 99.11% | 86.33% | 85.97% | 92.89% | 92.90% | 81.67% | 77.15% |
| Unimodal-2 | 82.80% | 83.19% | 72.17% | 71.26% | 97.17% | 97.19% | 86.25% | 85.80% | 92.25% | 92.22% | 81.59% | 76.25% |
| Unimodal-3 | - | - | - | - | - | - | - | - | - | - | 81.84% | 75.77% |

Table 4 reports UDI performance when different single-modal branches are chosen as the anchor. From table 4, we can draw the following conclusions:

- The selection of the best anchor depends on the performance of the modality branch on the dataset. For CREMA-D, Unimodal-2 (video) produces higher performance than Unimodal-1 (audio) as an anchor, whereas for Kinetics-Sounds the Unimodal-1 (audio) branch is superior. This confirms that the most useful anchor is not a fixed modality branch across tasks but depends on which modality branch carries the task-relevant information for that dataset.

- The degree to which anchor choice matters varies on different datasets. In CGMNIST, the gap between anchors is large, reflecting that the color channel is largely redundant/noisy and the gray branch is clearly superior as an anchor. In contrast, UCF101 shows only a marginal difference between anchors, indicating that when two modalities provide similar, overlapping information (RGB and optical flow).

- Choosing the highest-performing branch as the anchor produces the best results, but even without an extra anchor-selection step UDI still yields competitive performance on many datasets. This further highlight the effectiveness of UDI which employ a decoupling-based directed information flow strategy.

## A.8   SUPPLEMENTARY FIGURE

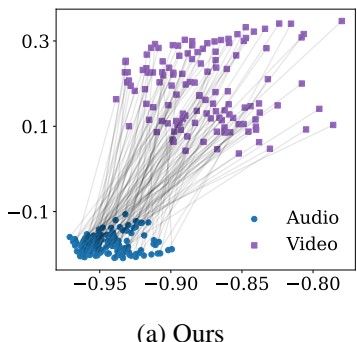
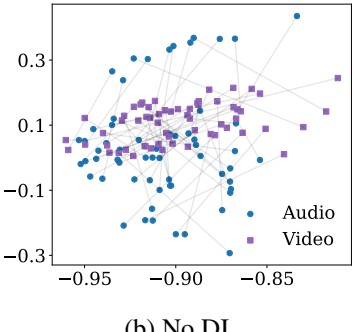

(a) Ours                    (b) No DI

Figure 5: Visualizations of the modality gap distance on the CREMA-D dataset.

