# OpenReview forum: "Balanced Multimodal Learning: An Unidirectional Dynamic Interaction Perspective"
_ICLR.cc/2026/Conference — ICLR 2026 Conference Withdrawn Submission_

### Official Review · Reviewer_Ui4c · 2025-10-17

**Soundness:** 3
**Presentation:** 3
**Contribution:** 1
**Rating:** 2
**Confidence:** 5

**Summary:**

This paper aims to address the modality imbalance problem in multimodal learning driving from the joint loss. They propose the Unidirectional Dynamic Interaction (UDI) strategy to train anchor modality independantly and distill the knowledge from anchor modality to other modalities. This framework can avoid the imbalance phenomenon. A consistency loss and a complementary loss are used to ensure the balance between the output consistency between anchor and follower modalities, and the complementary knowledge exploitation from follower modalities. Further,  a dynamic aware mechanism is introduced to keep the balance dynamically.

**Strengths:**

The motivation is sound and the proposed method is reasonable for me. The independant training of anchor modality can be used to avoid modality imbalance from joint loss, and the corresponding consistency and complementary losses are rational to be designed and applied here. For the experimental results, this paper uses datasets with several modality combinations, which is comprehensive and reasonable.

**Weaknesses:**

1. The motivation of this paper is similar with DI-MML [1], as well as the independant feature extractor training scheme, which makes this paper seem to lack innovation. Although the design of methods is not consistent, their cores are similar, and this paper also lacks comparative experiments with DI-MML.
2. In Figure 1, only one dataset is used for demonstration,. More datasets can be used for robust verification. Moreover, there lacks the explanation about why joint loss leads to modality imbalance.
3. In Section 2.2, the background about the application of MI in multimodal learning is insufficient.
4. For experimental results, the performance improvement of UDI is obvious only on CREMA-D and KS, and is very limited on other datasets. Does this indicate that this method is only superior on the audio-video dataset？
5. The unimodal performance is missed here, which should be listed to verify the effectiveness of addressing modality imbalance. And in Figure 1, I don't understand why the performance of follower modality (audio) in UDI is the same as that of Decoupled, with the participation of consistency and complementary losses. Shouldn't the performance of audio be better?
6. In Table 3, from the results, we can see that the adjustment of the coefficients has a limited impact on performance. Especially when the consistency is set to 1, it can achieve almost the same performance as UDI, even on the Food-101 which requires coefficients to be 0.5 as shown in Figure 3 (e). This weakens the necessity of dynamic adjustment.
7. Although the increased training cost of UDI has been explained in paper, I still believe that this method introduces too more cost, especially in the determination of anchor modality. Moreover, the training time comparison results should be illustrated.
8. The following methods are missing in comparison.

[1] Fan, Yunfeng, et al. "Detached and interactive multimodal learning." Proceedings of the 32nd ACM International Conference on Multimedia. 2024.
[2] Guo, Zirun, et al. "Classifier-guided gradient modulation for enhanced multimodal learning." Advances in Neural Information Processing Systems 37 (2024).
[3] Yang, Yang, et al. "Learning to rebalance multi-modal optimization by adaptively masking subnetworks." IEEE Transactions on Pattern Analysis and Machine Intelligence (2025).

**Questions:**

1. In UDI, whether the follower modality is trained from random initialization or fine-tuned from independant trained network?
2. I want to know why the authors choose the JS divergence in Eq.(4) but not other divergence losses such as simple KL or some metric learning losses?
3. In Figure 3 (f), the trimodal training is difficult to stabilize, and the two coefficients are also unstable. Is it due to the use of mini batch calculations or is this method inherently unstable in the trimodal field?

---

### Official Review · Reviewer_rBG4 · 2025-10-30

**Soundness:** 2
**Presentation:** 2
**Contribution:** 2
**Rating:** 4
**Confidence:** 4

**Summary:**

This paper proposes a training paradigm to address the prevalent issue of modality imbalance in multimodal learning. The authors introduce a Unidirectional Dynamic Interaction (UDI) strategy, which aims to proactively eliminate hidden competition among modalities. The core idea is to fully decouple the optimization process of different modalities during training: the modality branch with the best unimodal performance is first selected as the anchor, trained to convergence independently, and then frozen. The anchor’s learned predictions and feature representations are subsequently used to guide other modalities through unsupervised consistency and complementarity losses, establishing a directed information flow for balanced multimodal learning.

**Strengths:**

⦁	The paper proposes a proactive approach that fundamentally avoids modality competition, in contrast to prior reactive strategies relying on joint loss reweighting or gradient modulation.
⦁	The dynamic controller adaptively balances consistency and complementary terms, showing good generalization across tasks and datasets.

**Weaknesses:**

⦁	The paper does not sufficiently discuss how its approach differs from existing decoupled optimization methods such as Modality-Valuation [1], ReconBoost [2], MLA [3], or Remix [4]. Since these also decouple modality training to encourage sufficient unimodal learning, comparisons on unimodal performance in Figure 1 should include these baselines.
⦁	The proposed method requires independent pretraining for all modalities, which introduces significant computational overhead and may be far less efficient than joint optimization. The authors should post the training time or computation, and additionally compare with knowledge distillation–based methods or models that reuse pretrained unimodal weights.
⦁	The experimental section mentions that the dynamic adjustment strategy balances the complementary and consistency losses via the modality gap, but this claim lacks quantitative or theoretical clarification.
⦁	Nearly two-thirds of a page is devoted to the discussion and derivation of mutual information (MI), yet the formulation appears largely derived from existing works [5]. The paper should clarify its novelty in MI estimation or integration; otherwise, this section feels unnecessarily long and redundant.
⦁	Some recent strong baselines InfoReg[5] (CVPR2025), Remix[4] (ICML2025), DGL[6] (ICCV2025), AMSS[7] (TPAMI2025) are missing from the comparison
[1] Yake Wei, Ruoxuan Feng, Zihe Wang, and Di Hu. Enhancing multimodal cooperation via samplelevel modality valuation.
[2] Cong Hua, Qianqian Xu, Shilong Bao, Zhiyong Yang, and Qingming Huang. ReconBoost: Boosting can achieve modality reconcilement.
[3] Xiaohui Zhang, Jaehong Yoon, Mohit Bansal, and Huaxiu Yao. Multimodal representation learning by alternating unimodal adaptation.
[4] Xiaoyu Ma, Hao Chen, and Yongjian Deng. Improving Multimodal Learning Balance and Sufficiency through Data Remixing.
[5] Chengxiang Huang, Yake Wei, Zequn Yang, and Di Hu. Adaptive unimodal regulation for balanced multimodal information acquisition
[6] Shicai Wei, Chunbo Luo, and Yang Luo. Boosting Multimodal Learning via Disentangled Gradient Learning.
[7] Yang Yang, Hongpeng Pan, Qing-Yuan Jiang, Yi Xu, and Jinghui Tang. Learning to Rebalance Multi-Modal Optimization by Adaptively Masking Subnetworks.

**Questions:**

Refer to the weaknesses

---

### Official Review · Reviewer_wLSm · 2025-10-31

**Soundness:** 3
**Presentation:** 3
**Contribution:** 3
**Rating:** 4
**Confidence:** 5

**Summary:**

This paper introduces Unidirectional Dynamic Interaction (UDI), a novel training strategy designed to address the problem of modality imbalance in multimodal learning, where a dominant modality can suppress weaker ones during joint training. Instead of using a joint loss that encourages competition between modalities, UDI decouples their training. It first selects the best-performing modality (the anchor), trains it to convergence, and freezes it. Then, it uses the anchor's representations to guide the training of weaker (follower) modalities via a unified unsupervised loss. This creates a directed, non-competitive information flow. UDI replaces reactive joint-loss methods with a proactive, decoupled approach. This eliminates hidden competition from the start by sequentially training modalities. Extensive experiments on six benchmark datasets show that UDI outperforms a wide range of state-of-the-art baselines in handling modality imbalance.

**Strengths:**

1. Instead of proposing another improved technique for balancing a joint loss (e.g., better gradient modulation or loss weighting), the authors identify the joint loss itself as the root cause of the problem.  Their solution, Unidirectional Dynamic Interaction (UDI), is highly original because it abandons joint optimization entirely in favor of a proactive, sequential, and decoupled training paradigm.

2. The experimental design is thorough and robust. The authors validate their method against a comprehensive suite of 14 recent baselines on six diverse datasets, covering various modality types (audio-video, image-text, etc.).

**Weaknesses:**

1. Due to the selection strategy of anchor modality, the proposed method is not practical in real multimodal learning applications. For example,

2. The proposed method is computationally intensive.

**Questions:**

1. In Appendix A1, the authors state that "$\log p(y)$ is independent of $x$," and consequently treat $\log p(f^m)$ as independent of $f^a$. This assertion may be problematic. As established, $f^m$ and $f^a$ are likely dependent because, despite originating from different modalities ($x^m$, $x^a$), their source modalities are both connected to the same label $y$. If this dependency holds, it would undermine the proof. Therefore, the authors should clarify their assumption of independence between these features.


2. How does the computational efficiency (running time) of the proposed method compare to that of state-of-the-art baselines?


3. It is unclear how the proposed method could be implemented when there are more than two modalities, such as 3 or 4.


4. The experimental comparison may not be entirely fair, as the proposed method employs significantly more iterative updates than the baselines. It would be better that the authors could provide further explanation to justify this design choice and discuss its impact on the fairness of the comparison.

---

### Official Review · Reviewer_jLNT · 2025-11-01

**Soundness:** 3
**Presentation:** 2
**Contribution:** 2
**Rating:** 4
**Confidence:** 2

**Summary:**

This paper addresses the long-standing modality imbalance problem in multimodal learning and proposes a novel training paradigm called Unidirectional Dynamic Interaction (UDI). Unlike previous approaches that rely on joint loss functions to reactively balance modalities, UDI adopts a sequential decoupling strategy: each modality is first trained independently, and the best-performing one is selected as the anchor modality to guide others via unsupervised consistency and complementarity losses. Furthermore, a Dynamic Aware Mechanism adaptively adjusts these two loss weights based on gradient alignment, enabling task-adaptive and balanced cross-modal interaction. Experiments on six multimodal benchmarks demonstrate consistent improvements in both accuracy and F1 scores over thirteen state-of-the-art baselines. Overall, this work introduces a proactive and interpretable framework for mitigating modality imbalance, providing a fresh optimization perspective for future multimodal fusion research.

**Strengths:**

- The paper revisits the long-standing problem of modality imbalance and proposes a proactive training framework—Unidirectional Dynamic Interaction (UDI). This approach eliminates the dependency on conventional joint-loss–based balancing schemes and introduces a new optimization perspective for multimodal fusion, demonstrating strong methodological originality.
- The authors conduct systematic experiments on six representative multimodal datasets covering diverse modality combinations, including audio–video, image–text, and grayscale–color pairs. Across thirteen state-of-the-art baselines, UDI consistently achieves the best or near-best performance in terms of both accuracy and F1 score, confirming its broad applicability and robustness.
- The proposed UDI framework is clearly defined and theoretically grounded, consisting of three key components—anchor modality learning, follower modality adaptation, and a gradient-alignment–based dynamic weighting controller. Each module is conceptually coherent and mathematically well formulated, providing interpretability and transparency to the overall optimization process.

**Weaknesses:**

- The proposed method requires training each modality independently to select the anchor modality before proceeding to the subsequent guidance stage. This makes the overall training process relatively complex and computationally expensive. However, the paper does not provide quantitative comparisons with baseline methods in terms of training time, computational cost, or convergence speed, making it difficult to fully assess the scalability and practical applicability of the approach in large-scale tasks.
- The paper adopts a heuristic rule that selects the modality with the best validation performance as the anchor modality. However, if weaker modalities contain critical semantic information, this strategy may overlook important signals or even introduce bias. The work does not propose a more flexible or data-driven mechanism for anchor modality selection.
- The information flow in UDI is unidirectional, propagating only from the anchor modality to the other modalities. While this structure helps prevent competition between modalities, it may also hinder weaker modalities from correcting the biases of the anchor, thereby weakening the bidirectional complementarity and collaborative representation capabilities of the multimodal system.
- Although the experiments cover multiple multimodal datasets (such as audio–video, image–text, and grayscale–color combinations), these tasks are relatively small in scale. The paper has not yet validated the scalability and stability of the proposed method in large-scale multimodal pretraining or cross-domain scenarios.

**Questions:**

See weaknesses.

---

### Note · Authors · 2025-11-15

I have read and agree with the venue's withdrawal policy on behalf of myself and my co-authors.